# Structural Analysis of the Black-Legged Tick Saliva Protein Salp15

**DOI:** 10.3390/ijms23063134

**Published:** 2022-03-15

**Authors:** Belén Chaves-Arquero, Cecilia Persson, Nekane Merino, Julen Tomás-Cortazar, Adriana L. Rojas, Juan Anguita, Francisco J. Blanco

**Affiliations:** 1Centro de Investigaciones Biológicas, CIB-CSIC, Ramiro de Maeztu 9, 28040 Madrid, Spain; belen.chaves@cib.csic.es; 2Swedish NMR Centre, Medicinaregatan 5c, 41390 Goteborg, Sweden; cecilia.persson@nmr.gu.se; 3CIC bioGUNE, Parque Científico y Tecnológico de Bizkaia, 48160 Derio, Spain; nmerino@cicbiogune.es (N.M.); julen.tomascortazar@ucd.ie (J.T.-C.); arojas@cicbiogune.es (A.L.R.); janguita@cicbiogune.es (J.A.)

**Keywords:** Salp15, tick protein, protein structure, NMR

## Abstract

Salp15 is one of the proteins in the saliva of the tick *Ixodes scapularis*. Together with other biomolecules injected into the mammalian host at the biting site, it helps the tick to sustain its blood meal for days. Salp15 interferes with the cellular immune response of the mammalian host by inhibiting the activation of CD4^+^ T-lymphocytes. This function is co-opted by pathogens that use the tick as a vector and invade the host when the tick bites, such as *Borrelia burgdorferi*, the causative agent of Lyme borreliosis. Because of the immunity-suppressing role of Salp15, it has been proposed as a candidate for therapeutic applications in disorders of the immune system. The protein is produced as a 135-residue long polypeptide and secreted without its N-terminal signal 1–21 sequence. Detailed structural studies on Salp15 are lacking because of the difficulty in producing large amounts of the folded protein. We report the production of Salp15 and its structural analysis by NMR. The protein is monomeric and contains a flexible N-terminal region followed by a folded domain with mixed α + β secondary structures. Our results are consistent with a three-dimensional structural model derived from AlphaFold, which predicts the formation of three disulfide bridges and a free C-terminal cysteine.

## 1. Introduction

During feeding, ticks inject saliva into the host. Tick saliva is a cocktail of molecules with several pharmacological functions that, for example, inhibit pain, prevent blood clotting, and suppress the immune system response, thus facilitating the tick to feed unperturbed for long periods of time [1]. The protein Salp15 was identified in the saliva of the hard tick *I. scapularis* as one of 14 proteins recognized by sera of rabbits that had been fed upon by ticks [2]. Salp15 was later found to inhibit the activation of CD4^+^ T-cells [3]. The binding was mapped to the C-terminal 20-residue long region of Salp15 and the ectopic D1-D2 immunoglobulin domains of CD4 [4]. These binding studies were conducted with Salp15 recombinantly expressed in Drosophila S2 cells, yielding a glycosylated protein that binds to lectins specific for mannose and galactose structures [5]. The nature of the glycans in Salp15 secreted by ticks is unknown, but it is assumed to be similar to that of the protein produced by Drosophila S2 cells. Salp15 glycosylation is not necessary for its interaction with CD4 [3], but it is necessary for interaction with the carbohydrate-binding protein DC-SIGN, as deglycosylation of recombinant Salp15 by N-glycosidase F abrogated the interaction [5]. The Salp15 interaction with DC-SIGN occurs on the surface of dendritic cells and inhibits the production of pro-inflammatory cytokines. Salp15 also binds to *B. burgdorferi* outer surface protein, which protects the bacterium from antibody-mediated killing by the host [6]. The expression of Salp15 in the tick salivary gland is enhanced in ticks infected with *B. burgdorferi*, indicating that the manipulation of the tick saliva composition by the presence of the bacterium facilitates its survival in the host [1]. Because of the immunosuppressive properties of Salp15, it has been proposed as a candidate for therapeutic applications in disorders like autoimmune diseases.

Small-angle X-ray scattering (SAXS) on the recombinant Salp15 protein produced in Drosophila S2 cells (with a non-native 23-residue long tag at its C-terminus), showed an elongated globular shape [7], which is consistent with a monomeric glycosylated protein of about 25 kDa apparent molar mass (as seen in SDS-PAGE). Further structural analysis was hampered by the low amount of pure folded protein that could be produced from this source. Production of Salp15 in *E. coli* results in unfolded protein, as judged by the poor dispersion of backbone amide signals in ^1^H-^15^N NMR spectra [8]. We present here the production of milligram amounts of folded Salp15 suitable for structural analysis. Based on our previous experience with a cysteine-rich protease inhibitor from another tick species, we used a redox buffer to refold the protein produced in insoluble form by the bacteria and studied it by NMR and other techniques. The refolded protein is monomeric, contains a flexible N-terminal region, three disulfide bridges, a C-terminal reduced cysteine, and secondary structure elements consistent with the three-dimensional structure predicted by AlphaFold [9]. The availability of Salp15 in large quantities and the assignment of its NMR spectrum allows further investigation of its interaction with other molecules and explore its potential as a therapeutic agent.

## 2. Results

The production of the Salp15 protein (without the signal peptide) by the bacterial cells was very high but in the form of insoluble material. Part of the insoluble protein could be dissolved in denaturing buffer containing urea, and part of the solubilized denatured protein could be refolded by dilution in a redox buffer and purified by chromatography, as described in the methods section. The final yield of concentrated pure folded protein (Figure 1A) was 1.7 mg per liter of bacterial cells grown in an auto-induction medium.

SEC-MALS analysis yielded a molar mass of 12.5 kDa, corresponding to monomeric Salp15 (Figure 1B), which has a molar mass of 12.7 kDa calculated from the amino acid sequence. The circular dichroism spectrum of the protein (Figure 1C) indicated a mixture of secondary structures. Thermal denaturation followed by CD showed that the structure is stable, with an apparent mid-point denaturation temperature of approximately 60 °C (Figure 1D).

Initial investigation by NMR in the presence of 1 mM DTT showed that the protein became progressively denatured over a period of days, presumably due to the reduction of cysteines involved in intramolecular disulfide bridges. Therefore, subsequent protein preparation and analysis by NMR were done in the absence of DTT. The assignment of the backbone amide NMR signals was obtained for 101 of the 107 non-proline Salp15 residues (Figure 1E). This assignment has been deposited in the Biological Magnetic Resonance Data Bank (BMRB) as entry 51281. The small heteronuclear NOE values for residues 23–44 indicated that the N-terminal region of Salp15 is highly flexible (Figure 2). The central residues 72–79 and the C-terminal residues 134–135 (with heteronuclear NOE values smaller than 0.5) are also more flexible than other more rigid regions of the chain (Figure 2).

The analysis of the assigned chemical shifts identified a long α-helix spanning residues H54-S68, and two β-strands spanning residues T94-H99 and N104-N108 (Figure 2). Therefore, the global fold of Salp15 likely consists of a disordered N-terminal region and a globular domain with an α-helix, a β-sheet, and regions with non-regular structure. The three-dimensional structure of Salp15 is likely stabilized by three disulfide bridges involving six of the seven cysteine residues, whose ^13^C_α_ and ^13^C_β_ chemical shifts correspond to the oxidized form of cysteine [10]. Only C135, the C-terminal residue, is reduced, according to its ^13^C chemical shifts (Table 1). From the chemical shifts alone, however, we cannot derive the exact pairing of the cysteines in disulfide bridges.

The recently described tool for protein structure prediction AlphaFold allows obtaining three-dimensional models of proteins with unprecedented accuracy [9]. Whole proteomes have been fed through its algorithm yielding the corresponding structures for several organisms in the AlphaFold database (https://alphafold.ebi.ac.uk/, accessed on 22 January 2022). For those proteins not yet in this database (like Salp15), it is possible to predict their structure using a slightly simplified version of AlphaFold that does not consider existing structural templates using the Colab server (see methods). AlphaFold provides a quantitative measure of the prediction confidence at the residue level, and the regions with low-confidence values have been found to correlate with predicted disordered regions in the human proteome [11]. The prediction for Salp15 is highly consistent with our experimental data (Figure 2 and Figure 3).

The structural prediction for the N-terminal region is of very low confidence, in agreement with its highly flexible nature. AlphaFold predicts with high confidence an α-helix between residues H54 and S68, and an antiparallel β-sheet with three strands spanning residues D81–89, T94-H99, and V103-N108 (Figure 3). This distribution of secondary structure is very similar to that derived from NMR chemical shifts. The major difference is the length of the first β-strand, which is longer in the AlphaFold structure. AlphaFold predicted that the C-terminal C135 is the only cysteine not involved in disulfide bridge formation, in agreement with the NMR data. The other six cysteines are predicted to form the following disulfide bridges: C65-C97, C93-C122, and C115-C128. This pairing’s direct experimental verification could be achieved by mass spectrometry analysis of proteolytic fragments.

The three-dimensional structure of Salp15 predicted by AlphaFold is highly likely to represent the protein structure. The protein structure comparison server Dali [12] identifies the secreted type 1 cystatin from the helmint parasite *Fasciola hepatica* as the most similar single-domain protein structure in the Protein Data Bank (RMSD for the α-carbons is 3.1 Å, excluding the N-terminal flexible region. The crystal structure of this protein (PDB entry 6I1M) shows a fourth β-strand (Figure 3), but the overall fold is very similar despite a very low sequence identity (9%). The cystatin contains two disulfide bridges but their location in the structure is very different from the disulfide bridges in Salp15.

## 3. Discussion

Because of its immunosuppressive effect on the host, Salp15 plays a major role in sustained tick blood-feeding and in the transmission of tick-borne pathogens. Salp15 is one of the 19 *I. scapularis* salivary protein genes included in a recent mRNA vaccine that induces tick resistance and prevents transmission of the Lyme disease agent in guinea pigs [13]. The immunosuppressive properties of Salp15 make it also an interesting candidate for therapeutic applications in immune disorders [14]. Understanding Salp15 function and its potential development as a therapeutic agent will benefit from the knowledge of its structure, but this knowledge is hampered by the difficulty in the production of pure folded protein in sufficient amount.

Protein production in bacterial *E. coli* cultures is very convenient because they grow quickly in cheap media to high cell density. However, cysteine-rich proteins are generally difficult to produce because the reducing nature of the bacterial cytoplasm does not favor the formation of disulfide bridges. This problem can sometimes be circumvented by expressing the protein in the oxidizing bacterial periplasm, but at the cost of low yields [15]. The formation of the native set of disulfide bonds is the end result of a process involving covalent reactions such as cysteine oxidation into cystine, cystine reduction, and disulfide bridge isomerization/reshuffling [16], as found in the early days of protein folding studies with bovine pancreatic trypsin inhibitor as a model protein [17,18]. For this reason, it is helpful to co-express sulfhydryl oxidases and disulfide bond isomerases for the efficient production of folded disulfide-rich proteins in the bacterial cytoplasm [19]. 

A simpler method can be used when the protein is highly expressed but in the form of insoluble material. The protein can then be solubilized in a denaturing and reducing buffer and refolded in a redox buffer that facilitates disulfide bridge formation. This approach has been successfully applied for protease inhibitors, with yields of the folded protein, with native disulfide bridges, high enough for structural studies [20]. We have used this strategy to produce Salp15 in a simple way to investigate its structure by NMR and other complementary techniques. The finding that the N-terminal residues were highly flexible led us to design two shorter versions of the gene (coding for amino acids 28–135 and 40–135), that could produce proteins suitable for crystallization. However, these N-terminal deletion versions of Salp15 were not well folded based on their NMR spectra. The functional role of the long flexible N-terminus is unknown, but it could be necessary to facilitate folding and prevent aggregation [21].

The three disulfide bridges, according to the AlphaFold model, stabilize a globular but elongated structure with an α-helix and a three-stranded antiparallel β-sheet. The loop between the helix and the first β-strand is also flexible, as is the C-terminal end, which contains the only reduced cysteine of Salp15. Using a panel of immobilized 11 overlapping 20-residue long fragments of Salp15, it was found that the C-terminal 20 residues were necessary for binding to the soluble CD4 domains D1-D2 [4], and that the peptide caused a dose-dependent inhibition of CD4^+^ T-cells, thus recapitulating the immunosuppressive effect of Salp15 [22]. This is somewhat unexpected in light of the structure of Salp15, because this C-terminal peptide contains two cysteines involved in disulfide bridges with other cysteines not present in the peptide. It is highly unlikely that the isolated peptide maintains the same folded structure as in the whole protein, but it could be that the C-terminal primary structure, not the tertiary structure, is the determinant factor for CD4 binding. In fact, flow cytometry using fluorescently labeled proteins showed that a Salp15 deletion mutant lacking the C-terminal 15 residues is able to bind to purified CD4^+^ T-cells, albeit with less efficiency than full-length Salp15 [14]. Further studies on Salp15 interactions will be necessary to clarify this and other aspects of Salp15 structure-function relationship, which will be facilitated by the availability of the assigned NMR spectrum.

## 4. Materials and Methods

*Protein Expression and Purification.* The gene to produce *Ixodes scapularis* Salp15 fragment 23–135 (the Uniprot Q95WZ4 sequence without the signal peptide) was amplified by PCR and cloned into the plasmid named pHIS-Parallel2 using NcoI and SalI restriction sites. *E. coli* cells BL21(DE3) transformed with this plasmid and grown in LB medium were selected with Ampicilin. The produced polypeptide chain includes an N-terminal His_6_-tag followed by a tobacco etch virus protease (TEV) site. After proteolysis, a protein with the non-native GAMG sequence preceding the first native residue (E23) is obtained. The cells were grown in auto-induction medium ZYP-5052 [23] for natural isotope abundance, in auto-induction medium P-5052 [24] for uniform ^15^N-enrichment (containing ^15^N-NH_4_Cl at 98% enrichment), and in minimal medium M9 for uniform ^13^C,^15^N-enrichment [25], with ^15^NH_4_Cl (98% enrichment) and [^13^C_6_]-D-glucose (99% enrichment) as the only nitrogen and carbon sources. ZYP-5052 was inoculated with cells pelleted from a 20-fold smaller culture grown overnight in the same medium at 37 °C. P-5052 was inoculated with the pellet from a 20-fold smaller culture grown overnight in PA-0.5G [24] at 37 °C. The M9 medium was inoculated with the cells pelleted from a 3-fold larger culture grown in LB medium at 37 °C until OD_600nm_ = 1. The auto-induction media were grown at 37 °C for 3 h and then at 20 °C for approximately 20 h. The inoculated M9 medium (with OD_600nm_ = 3) was grown at 37 °C for 0.5 h, induced with 1 mM IPTG, and grown for approximately 20 h at 20 °C.

The cells were harvested by centrifugation and resuspended in lysis buffer, which contains 20 mM Tris pH 8.0, 300 mM NaCl, 1 mM DTT, and *Complete EDTA-free* protease inhibitors (Merck, Darmstadt, Germany). The suspension was frozen in liquid nitrogen and stored at −80 °C. After thawing, small amounts of lysozyme and DNAse were added and the cell suspension was sonicated and ultracentrifuged at 4 °C. The Salp15 protein was found predominantly in the insoluble fraction, as seen by SDS-PAGE.

The pellets were washed by resuspension in lysis buffer with 2% (*v*/*v*) Triton X-100, then washed by resuspension in lysis buffer, and then partially solubilized in lysis buffer with 7 M urea. Most of the protein was present in the soluble fraction (typically 6 mL for each liter of bacterial culture), and was refolded by a 50-fold dilution drop-by-drop in lysis buffer containing 2 mM cistine and 4 mM cysteine at 4 °C and with a gentle stirring. After about 20 h at 4 °C, the sample showed partial precipitation and was clarified by filtration before loading into a 5 mL His-Trap column at 4 °C using a peristaltic pump. The column was then washed with 5 column volumes of the lysis buffer and with 5 column volumes of buffer with 50 mM imidazol. The protein was eluted with 7 column volumes of buffer containing 500 mM imidazol. Salp15 containing fractions were pooled and the total amount of protein was estimated by ultraviolet absorbance (using the extinction coefficient of 9315 M^−1^ cm^−1^ at 280 nm computed from the amino acid sequence of salp15). An aliquot of a TEV protease stock (at approximately 1 g/L in 50 mM sodium phosphate buffer at pH 8.0 with 200 mM NaCl, 1 mM DTT, and 10% glycerol) was added to the pooled fractions (approximately 1 mg of protease for every 30 mg of salp15) and incubated at 4 °C in a dialysis tubing (3000 kDa cut-off) against lysis buffer at pH 7.5 for about 20 h (in order to remove the imidazol at the same time that the fusion protein was cleaved). The sample was filtered and loaded into the His-Trap column. Part of the cleaved protein was retained in the column (and was later found to be unfolded protein), but only the cleaved Salp15 contained in the flowthrough was further concentrated and chromatographed through a Superdex 75 16/60 equilibrated in lysis buffer at pH 7.5. The protein eluted in two peaks of similar height and width centered at 73.7 and 80.1 mL, which contained the same polypeptide chain (according to MALDI-TOF and reducing SDS-PAGE). The second peak yielded a ^1^H-^15^N-HSQC NMR spectrum with sharp and dispersed backbone amide signals, typical of folded proteins. The first peak showed a set of signals that were broader and non-dispersed, with backbone amide signals between 7.7 and 8.6 ppm. The fractions of the second peak were used for further characterization by circular dichroism, SEC-MALS and NMR. The final yield was approximately 1.7 mg per liter of bacterial culture.

*Size exclusion chromatography-multi angle light scattering (SEC-MALS).* These experiments were performed at 25 °C using a Superdex 75 10/300 GL column (Cytiva, Marlborough, MA, USA) connected to a DAWN-HELEOS light scattering detector and an Optilab rEX differential refractive index detector (Wyatt Technology, Santa Barbara, CA, USA). The column was equilibrated and run in 20 mM Tris-HCl at pH 7.5 with 300 mM NaCl, 0.5 mM TCEP, and 0.03% NaN_3_ (0.1 µm filtered), and the SEC-MALS system was calibrated with a sample of Bovine Serum Albumin (BSA) at 1 g/L in the same buffer. A sample of 100 µL of Salp15 protein at 1.9 g/L was injected and chromatographed at 0.5 mL/min. Data acquisition and analysis employed ASTRA software (Wyatt Technology, Santa Barbara, CA, USA). Based on numerous measurements on BSA samples at 1 g/L under the same or similar conditions, we estimate that the experimental error in molar mass is around 5%.

*Circular dichroism (CD)*. These measurements were performed on a Jasco J-815 spectropolarimeter (JASCO, Tokyo, Japan) at 25 °C. The spectra were the average of 5 scans, recorded using a 0.1 mm path length quartz cuvette on a 30 μM Salp15 protein sample in 20 mM Tris-HCl at pH 7.5 with 300 mM NaCl, 1 mM DTT. Thermal denaturation was measured in a 2-mm path length cuvette closed with a teflon cap on the same protein stock solution. Temperature was increased at a rate of 1 °C/min from 20 °C to 95 °C, and the ellipticity at 222 nm was recorded at intervals of 1 °C with a 4-nm bandwidth and a response of 32 s.

*NMR samples and spectroscopy*. To prepare samples for the assignment of the NMR spectrum and ^15^N-relaxation measurements, folded Salp15 from the gel filtration chromatography was further purified by reverse phase chromatography on a Jupiter C_18_ 250 × 21 mm column (Phenomenex, Torrance, CA, USA). The protein was eluted with a gradient from 0 to 100% of 90% acetonitrile in water with 0.1% TFA, frozen in liquid nitrogen and lyophilized. The powder was reconstituted into 10 mM sodium phosphate at pH 5.5 with 50 mM NaCl, 7% (*v*/*v*) ^2^H_2_O, 0.01 NaN_3_ and 30 μM DSS by cycles of dilution and concentration by ultrafiltration until a volume of approximately 120 μL, which was transferred to a 3 mm NMR samples tube.

NMR experiments were performed at 298 K on a Bruker AVANCE III HD 800 spectrometer (Billerica, MA, USA) equipped with a cryoprobe. Sequence-specific polypeptide backbone chemical shift assignments were made on a 0.14 mM sample using the following 3D experiments: HNCO, HNCACO, HNCA, HNCOCA, HNCOCACB and HNCACB. TopSpin (Bruker, Billerica, MA, USA) and CCPN [26] were used to process and analyze the data. Chemical shifts were referenced to DSS used as an internal reference for ^1^H, and were calculated for ^15^N and ^13^C [27]. Secondary structural identification from chemical shifts was made with TALOS+ [28]. {^1^H}-^15^N NOE spectra [29] were acquired in the interleaved mode on a 1 mM uniformly ^15^N labeled sample, with proton saturation during the 5 s relaxation delay. Heteronuclear NOEs were calculated from the ratio of cross-peak intensities in spectra collected with and without amide proton saturation during the recycle delay. The error was calculated from the estimated noise intensity in the spectra.

*Three-dimensional structure prediction and analysis.* The simplified version of AlphaFold was run on the Salp15 sequence 23–135 using the Colab server (https://colab.research.google.com/github/deepmind/alphafold/blob/main/notebooks/AlphaFold.ipynb, accessed on 22 January 2022). Secondary structure identification was done with the DSSP algorithm [30], and the structural model was visualized with PyMol (Schrödinger, New York, NY, USA).

## 5. Conclusions

Salp15 is a monomeric protein that can be efficiently produced from *E. coli* cultures in its folded form. A long and disordered N-terminal region is followed by a globular domain with an α + β fold, which is stabilized by three disulfide bridges. The assignment of the NMR spectrum of Salp15 will facilitate studying the molecular recognition of Salp15 by its receptors.

## Figures and Tables

**Figure 1 ijms-23-03134-f001:**
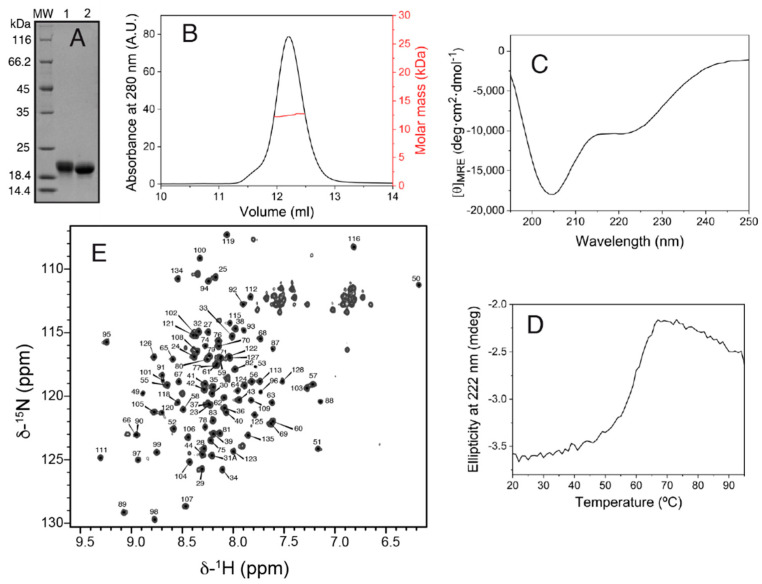
Structural characterization of Salp15. (**A**) Reducing SDS-PAGE of purified Salp15 in the unfolded form (lane 1, preparative gel filtration elution peak at 73.7 mL) and in the folded form (lane 2, elution peak at 80.1 mL). (**B**) SEC-MALS analysis of the folded Salp15. (**C**) Far-UV circular dichroism spectrum of the folded Salp15. (**D**) Thermal denaturation of the folded Salp15 followed by the change in the ellipticity at 222 nm. (**E**) ^1^H-^15^N HSQC NMR spectrum of Salp15 with the residue-specific assignment of the backbone amide resonances.

**Figure 2 ijms-23-03134-f002:**
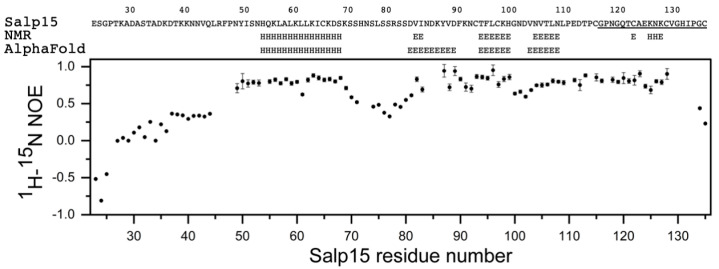
NMR analysis of Salp15. Salp15 sequence (residues 23–135), secondary structure identification from assigned chemical shifts using TALOS+ or identified in the AlphaFold model (H: α-helix, E: β-strand), and heteronuclear {^1^H}-^15^N NOEs. The C-terminal 20 residues that were found to be necessary for Salp15 binding to CD4 are underlined.

**Figure 3 ijms-23-03134-f003:**
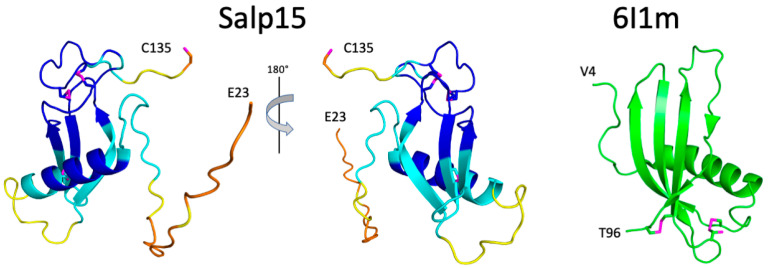
Structural model of Salp15 and similarity with Protein Data Bank entry 6I1m. (**Left**) AlphaFold predicted structure for residues 23–135 of Salp15. The color code of the main chain follows that used by AlphaFold to report the reliability of the models. Blue, cyan, yellow, and orange protein regions correspond to very high (LDDT > 90), high (90 > LDDT > 70), low (70 > LDDT > 50), and very low (LDDT < 50) model confidence, respectively. The side chains of the seven cysteine residues are shown as sticks, with the sulfur atoms in magenta. Only the C-terminal C135 side chain is reduced (according to NMR chemical shifts). The depicted secondary structure is that identified by DSSP. (**Right**) Crystal structure of secreted type 1 cystatin from *F. hepatica*. The side chains of the four cysteines are shown in sticks with the sulfur atom in magenta. C66 is modeled in two conformations (both are shown). The depicted secondary structure is that identified by PyMol. The N- and C-terminal residues are indicated for both proteins.

**Table 1 ijms-23-03134-t001:** Chemical shifts of the ^13^C_α_ and ^13^C_β_ resonances of cysteines in Salp15.

Salp15 Cysteine	^13^C_α_ (ppm)	^13^C_β_ (ppm)	Redox State ^a^
65	56.70	35.20	Oxidized
93	60.50	42.50	Oxidized
97	52.83	37.90	Oxidized
115	54.67	38.87	Oxidized
122	58.00	40.97	Oxidized
128	54.90	35.98	Oxidized
135	63.60	31.70	Reduced

^a^ Based on the statistical analysis of cysteine ^13^C chemical shifts in proteins [10], which yielded the following average and standard deviation values: ^13^C_α_ = 55.5 ± 2.5 ppm (oxidized Cys), ^13^C_α_ = 59.3 ± 3.2 ppm (reduced Cys), ^13^C_β_ = 40.7 ± 3.8 ppm (oxidized Cys), ^13^C_β_ = 28.4 ± 2.4 ppm (reduced Cys).

## Data Availability

The assignment of the NMR spectrum of Salp15 has been deposited in the Biological Magnetic Resonance Data Bank (BMRB) as entry 51281.

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
