# Peer review of "Structural Analysis of the Black-Legged Tick Saliva Protein Salp15"

_ijms, 2022, doi:10.3390/ijms23063134_

Round 1

Reviewer 1 Report

This paper characterizes Salp5 expressed by a purification method that takes into account disulfide bonds. Especially, the successful refolding of the insoluble cysteine-rich protein produced by the bacterium using redox buffer will be of great value to researchers working with this protein.

However, with the exception of NMR measuments, the characterization in this study was very simple and general, and has little scientific value. In addition, no functional analysis has been done to prove whether the refolded structure is in its natural state or not. 

The reduction/oxidization state of cysteine, the most important determinant of this protein structure, was predicted by NMR chemical shifts, but any experimental results for this prediction were not shown in the figures or tables in this paper. Instead, you used the results of structure prediction by AlphaFold to predict the position of disulfide bonds. Certainly, the prediction accuracy of AlphaFold is quite high, but the experimental evidence should be trusted more, and the positions of disulfide bonds should be demonstrated by experimental results such as mass spectrometry or structural analysis by NMR. 

You described that the predicted structure of Salp5 by AlphaFold is similar to type 1 cystatin from the helmint parasite Fasciola hepatica, but the structure comparison is not shown in the paper, although the RMSD values are described. It would be more helpful to compare the structure between Salp5 and cystatin in Figure 3.

Author Response

We have included a new table (Table 1) with the 13C chemical shifts of the seven cysteines of Salp15. The combined 13Calfa and 13Cbeta chemical shift values indicate that only C135 is reduced, in agreement with the AlphaFold model. We agree with the reviewer that experimental evidence should be trusted more than prediction. Regarding the exact pairing of the cysteine side chains in disulfide bridges, we acknowledge in the revised text that this could be achieved by mass spectrometry analysis of Salp15 proteolytic fragments.

We now show in Figure 3 a representation of the crystal structure of the cystatin to illustrate its similarity to the AlphaFold model of Salp15.

Reviewer 2 Report

The manuscript reports the production of the folded residues 23-135 of the Salp15 protein. This sequence was characterized by NMR. Its 3D modeling was done with the tools of the AlphaFold database. Only a qualitative comparison of the experimental data and the modeled structure is possible.

Not being an expert in this field, I cannot appreciate the practical importance of the presented results. However, the title of the manuscript should be appropriate to its content and not mislead the reader. The statement that the article determined the structure of the black-legged tick saliva protein Salp15 is incorrect.

Author Response

The title of the manuscript has been changed to follow the suggestion of the reviewer.

Reviewer 3 Report

The ms is on the structural characterization of protein Salp 15.

The task is not easy, however, different methods e.g. the not trivial NMR measurements and the slightly simplified version of AlphaFold  was extremely useful.

The ms (that is rather a communication) is sound and will be suitable for publication in Int. J. Mol. Sci after the following minor revisions:

  • The first part of the abstract is perhaps too lengthy. There is no need to give too much details on the role and utility of Salp 15.
  • Would not it be possible to show the amino acid sequence for the protein under discussion?
  • The conclusion part (summary) is missing. Pls point out the advantages and consequence sof the structure determination.

Author Response

Although perhaps not all the information in the abstract is necessary, it explains the relevance of Salp15, and its length is within the guidelines of the journal. Therefore, we prefer to keep it as it is.

The amino acid sequence of the protein that has been produced and characterized (Salp15 residues 23-135) is shown in Figure 3. The amino acid sequence of the protein including the N-terminal signal sequence can be found in the Uniprot entry Q95WZ4, as indicated in line 221.

A conclusion section has been added, as suggested by the reviewer.

Round 2

Reviewer 1 Report

I appreciate your honesty in responding to my first comment.

One character amino acid name D81-89 in line 134 on page 4 was missing.

The "structural analysis" in the new title is misleading since the structure was predicted but not analyzed in this paper. The term"structural analysis" is generally used to determine 3D structure, therefore it should be changed to a different term.